# The Effect of Acaricide Control of the Two-Spotted Spider Mite *Tetranychus urticae* Koch on the Cultivation of Sugar Beet (*Beta vulgaris* L.) and on the Size and Quality of the Yield

Jan Bocianowski [1,*] , Magdalena Jakubowska [2] , Daniel Zawada [3] and Renata Dobosz [4]

[1]  Department of Mathematical and Statistical Methods, Poznań University of Life Sciences, Wojska Polskiego 28, 60-637 Poznań, Poland
[2]  Department of Monitoring and Signalling of Agrophages, Institute of Plant Protection-National Research Institute, Węgorka 20, 60-318 Poznań, Poland
[3]  Sumi Agro Poland, Bonifraterska 17, 00-203 Warszawa, Poland
[4]  Department of Entomology and Animal Pests, Institute of Plant Protection-National Research Institute, Węgorka 20, 60-318 Poznań, Poland
*   Correspondence: jan.bocianowski@up.poznan.pl

**Abstract:** Field experiments (in the 2019–2021) were carried out at the Department of Field Experimentation of the Institute of Plant Protection—National Research Institute in Winna Góra, the purpose of which was to test the insecticidal and acaricidal effectiveness of sugar beet cultivation protection against *Tetranychus urticae* and to assess its impact on the size and quality of the sugar beet crop. In the experiment, the following acaricides were used: spirodiclofen—240 g—22.11%, mixture of hexythiazox—250 g—23.15% and fenpyroximate—51.2 g—5.02% and insecto-acaricide paraffin oil—770 g $L^{-1}$ (89.6%) and abamectine—18 g—1.88%. The controls were plants left without chemical protection. The plants were sprayed when ten mobile individuals/two spotted spider mites appeared on the leaves. Chemical treatments were carried out in the full growing season in the phase of leaf rosette formation (July–August). In the second half of October, the plant density (PD) in the field was estimated. Parameters characterizing the size and quality of the crop were calculated: sugar beet yield (SBY), biological sugar yield (BSY), pure sugar yield (PSY), sugar content (SC), refined of sugar content (RSC), the yield of preferential sugar (YPS), recoverable sugar (RS), potassium molasses (PM), sodium molasses (SM), α-amino nitrogen (α-AN), alkalinity factor (AF) and standard molasses losses (SML). The years were statistically significantly different for all 13 traits. Statistical differences were observed in the mean values of the observed parameters in these years, except for α-amino nitrogen (α-AN) and alkalinity factor (AF). The mean values of SBY, biological sugar yield (BSY), pure sugar yield (PSY) and sodium molasses (SM) differed depending on the type of protection applied. Positive correlations were observed for 28 pairs of traits, but negative statistically significant relationships were observed between 11 pairs of traits. The first two canonical variates accounted for 85.49% of the total variability between the individual combinations. The significant positive relationship with the first canonical variate was found for PD, BSY, PSY, SC, RSC, YPS, but negative for SM. The $CV_2$ was negatively correlated with: SBY, BSY, PSY, RS, PM, SM, α-AN and SML. The greatest variation in terms of all the 13 traits jointly was found for Vertigo 018 EC in 2020 and Vertigo 018 EC in 2021. The greatest similarity was found between control in 2019 and Ortus 05 SC in 2019.

**Keywords:** insecticidal protection; yield quality; efficiency; canonical variance analysis; Mahalanobis distances; Pearson's linear correlation

## 1. Introduction

In addition to sugar cane (*Saccharum* L.), sugar beet (*Beta vulgaris* L.) is the second most economically important plant grown for obtaining sugar. The main area of sugar beet production is Europe, concentrating 68.3% of the total sugar beet production in the

world in 2010–2020 [1]. During this period, beet production increased, on average from slightly over 228 to less than 253 million tons per year [1], with an observed decrease in the acreage allocated to this cultivation. On a global scale, the main producers of sugar beet from 2010–2020 were the Russian Federation, France and the USA [1].

Over the past five years, in Europe (EU, UK and Switzerland), an average of 1.5 million ha has been allocated to the cultivation of sugar beet. In the period 2020/2021, the acreage occupied by this crop was 2.5% lower than in 2017 and the leading countries at that time were France (370,000 ha), Germany (351,000 ha) and Poland (252,000 ha) [2]. In connection with the reduction of the sugar beet cultivation area and as a result of environmental and economic factors, a decrease in sugar beet production volume was observed by an average of about 11%, mainly in Spain, Croatia and Italy, with a simultaneous increase in the area of Scandinavian countries: in Denmark it increased by 14%, in Finland by 7% and in Sweden by 7%. The crop harvested in the period 2020/2021 was estimated at 14.2 million tons. The average sugar yield was estimated at 11.5 t ha$^{-1}$ in the last five years. On average, over the last five years, the sugar content of sugar beet produced in the EU-27 has been estimated at 17%. There was also a decrease in the sugar content in the yield from 16.5% in the 2019/2020 season to 15.9% in the 2020/2021 period [2].

The variability of the root yield and the percentage of sugar in the roots depend primarily on weather conditions. The summer period (July–September) is the most critical, due to water needs for the development of sugar beet. Appropriate intensity of solar radiation, distribution and sum of rainfall, as well as optimal air and soil temperatures contribute to a high level of plant emergence and their proper development. This results in a full plant density in plantations. Optimal meteorological conditions also ensure the proper course of photosynthesis and influence the accumulation of sugars in the plant [3–5]. Sugar beet is a plant with high soil requirements, preferring humus soils in good condition with a neutral or alkaline pH. This plant, under favorable conditions for development, gives large gains of green mass in a short period of time. In addition to the correct soil pH, sugar beet requires the application of nitrogen, phosphorus, magnesium and potassium to the soil and to supplement the necessary microelement—boron [6–8]. Excessive amounts of these elements, deficiencies or disturbances in their correct proportions in the soil lead to abnormalities in the development of sugar beet plants and to a reduction in the amount of sugar produced and deterioration of its quality, mainly due to the formation of molasses [9–16]. The correct date of sowing [17] and harvesting [18] also determines the success of the size and quality of the crop.

In addition to proper agri-techniques, the cultivation of sugar beet also requires effective protection of beet crop against pests. During the growing season, sugar beet plants are exposed to bacterial Pseudomonas [19], viral: BNYVV [19,20] and fungal pathogens: *Erysiphe betae* (Vanha), *Cercospora beticola* Sacc., *Ramularia beticola* Fautrey and Lambotte, *Uromyces betae* (Bellynck), *Phoma betae* Frank. and *Verticillium* spp. [19,21,22]. Sugar beet plants are sensitive to the occurrence of negative effects of weed infestation, especially in the initial stages of plant vegetation [23].

Sugar beet plantations are also attacked by many soil and ground pests, including plant parasitic nematodes: sugar beet cyst nematodes *Heterodera schachtii* Schmidt, 1879 and *H. betae* Wouts, Rumperhorst and Sturha, 2001, stubby root and virus vector nematodes *Trichodorus* and *Paratrichodorus* spp., stem nematode *Ditylenchus dipsaci* Filipjev, 1936 [21,24–26]; insects: larvae of noctuid moths e.g., *Euxoa* spp., *Agrotis* spp., *Peridroma saucia* (Hübner), *Xestia c—nigrum* (L.), *Crymodes devastator* (Brace), *Feltia ducens* (Walker) in the USA; *Scrobipalpa ocellatella* Boyd, *Spodoptera* spp. and *Pseudaletia unipunctata* (Haworth) [21,23,27–32] and spider mites, among which the harmfulness to sugar beet has been proven for the two-spotted spider mite *T. urticae* Koch, 1836 [33–36].

*T. urticae* is a polyphagic spider mite, inhabiting many species of crops, including sugar beet. The negative effects of *T. urticae* foraging have been observed in Poland for many years on sugar beet plantations located in intensively cultivated regions in the central voivodships [34]. These are areas with favorable climatic conditions for the development

of the spider mite: dry springs, high temperatures in summer between 25–30 °C and a low rainfall of 0–200 mm.

Symptoms of damage to beetroot plants by *T. urticae* appear at the earliest at the edge of the fields and over time become visible throughout the area in the form of plant clusters with yellowed and turgor-free leaves. The feeding effects of the spider mite can be seen both on the upper and lower side of the leaves in the form of light spots. Intensive feeding of *T. urticae*, on the other hand, results in the appearance of small shiny spots arranged in the form of a mosaic on the surface of the leaves. Symptoms caused by spider mites are often confused with symptoms caused by drought, viruses, or non-specific symptoms triggered by nematode infestation. A symptom directly indicating the presence of a spider mite is a delicate spider web covering the underside of the leaves, where all juveniles, adults and eggs are found. The development of the *T. urticae* population during sugar beet vegetation leads to leaf deformation, plant wilting and even complete dieback. In the climatic conditions of Poland, *T. urticae* can range from four to six generations during the growing season. Under favorable temperature conditions, usually occurring in late spring and summer, it may take only eight days for a complete generation of spider mites to develop. Intensive foraging of *T. urticae* leads to a decrease in yield (20–50%) and the sugar content in the roots may be lower by up to 2% [33,35,36].

In order to limit the damage caused by the spider mite, a chemical method is currently used, in addition to micro- and macro-organisms (the natural properties of plants). Active substances representing numerous chemical groups limiting the behavior of spider mites and changing their metabolic ratios are used.

The aim of the study was to estimate the effectiveness of selected active substances with insecticidal and acaricidal activity in limiting the development of the hop two-spotted spider mite population in sugar beet plantations and to assess the effect of the active substance used for controlling the size and quality of the sugar beet crop.

## 2. Materials and Methods

### 2.1. Description of the Experimental Site. Characteristics of the Soil

Field experiments were carried out at the Department of Field Experimentation of the Institute of Plant Protection—National Research Institute in Winna Góra (52°12′17″ N, 17°26′48″ E), from 2019 to 2021. The experiment in each year was based on a one factorial randomized block design, where different combinations of acaricides and time of application were treated as one combination factor. Six variants were studies: five acaricides and one control—without chemical treatment. Insecticide protection against two-spotted spider mites (TSSM) *Tetranychus urticae* was applied after the optimal date of plant protection product was established and the threshold of economic harmfulness was exceeded. The chemical product should be used in sugar beet from the phase of complete coverage of inter-rows until the end of root growth (BBCH 39–49), bearing in mind the grace period, which is 28 days.

The number of repetitions was four and the total number of plots was 24. Each plot included six rows. The area of the plot for sowing was 13.5 m$^2$ (width—1.8 m, length—7.5 m). Sowing was performed with a precision seed drill. The number of plants per plot was 108; when sowing sugar beet seeds the distance in the row were 24.0 cm and with a row spacing of 45.0 cm. The number of rows in the plot was six. The mean final plant density was 86–90 sugar beet plants per plot. The entire area of field plots was around 350 m$^2$.

The soil type was: clay—34%, sand—14%, silt—52%) (soil classes IIIa, IIIb and Iva). The soil reaction was neutral as required for sugar beet, with medium phosphorus (P) content and high potassium (K) and magnesium (Mg) content. Winter wheat was grown as the fore-crop for sugar beet in all years of study. Plots were fertilized with nitrogen (120 kg N ha$^{-1}$). 60 kg N ha$^{-1}$ nitrogen dose was applied before sowing and in BBCH 14. One week before sowing, the soil was fertilized with P, dose 60 kg of P$_2$O$_5$ ha$^{-1}$, combined

with K. Weed, disease and other pest control was conducted in accordance with the plant protection recommendations of the Institute of Plant Protection—NIR (Poland).

The sugar beet variety Marynia was used in the field experiment. The sugar beet seed variety was treated with Tachigaren 70 WP fungicide [hymexazol—active ingredient (a.i.)—700 g kg$^{-1}$—70%] in dose 40 g per seeds unit ha$^{-1}$. The seeds were sown between April 4–9 with a sowing density of 1.02 seeding unit per ha$^{-1}$. Chemical treatments were performed after finding traces of mites (TSSM) feeding on leaves, after the threshold of economic harmfulness was exceeded and after taking into account the analyzed criteria. In sugar beet, the economical threshold TSSM is not known. In our field experiment, we took 10 mobile stages of spider mite per plants as a critical point for a plant protection product. Chemical treatments were carried out using a plot sprayer with the recommended amount of water of about 400–450 l ha$^{-1}$. For spraying, ejector, two-stream TeeJet nozzles were used with an average droplet. The spraying fluid had a pressure of 0.3 MPa. The experimental layout is presented in Table 1, including the total doses of applications and mite stages by acaricides and insecticides application (per 1 ha).

**Table 1.** Insecticide/acaricide treatment against *Tetranychus urticae* and criteria used for application.

| Insecticide/Acaricide | Variant | Active Ingredients | Criterium for Application | Mite Stage | Dose |
|---|---|---|---|---|---|
| Envidor 240 SC | (E) | Spirodiclofen—240 g—22.11% | feeding symptoms | moving stages and eggs | 0.4 l ha$^{-1}$ |
| Nissorun Strong 250 EC + Ortus 05 SC | (NO) | Heksytiazoks—250 g—23.15% | feeding symptoms | egg stage | 0.4 l ha$^{-1}$ |
| | | Fenpiroksymat—51.2 g—5.02% | | moving stages | 1.5 l ha$^{-1}$ |
| Ortus 05 SC | (O) | Fenpiroksymat—51.2 g—5.02% | feeding symptoms | moving stages | 1.5 l ha$^{-1}$ |
| Treol 770 EC | (T) | paraffin oil—770 g L$^{-1}$ (89.6%) acaricide | feeding symptoms | moving stages | 1.5 l/100 l ha$^{-1}$ |
| Vertigo 018 EC | (V) | Abamectine—18 g—1.88%) | feeding symptoms | moving stages | 0.75 l ha$^{-1}$ |
| Control | C | Pesticide free | | | |

### 2.2. Data Collection

The sugar beets were harvested in the second decade of October, in each year. During harvest, the plants were topped by hand on the three middle rows and the leaves were weighed. The roots were then counted, dug up and weighed. During the harvest, each plot was collected in accordance with the Polish Standard (PN-R-74452).

The roots were collected by hand from the six rows (10.8 m$^2$). Three middle rows in the plot were intended for harvesting. Qualitative and quantitative parameters were analyzed for 20 roots of sugar beet. The root samples were pulped in the Plant Breeding Station of the Kutno Sugar Beet Breeding Company in Śmiłów. In Straszków (the Kutno Sugar Beet Breeding Company, Kłodawa, Poland) on the automatic Venema technological line [37], the sugar content polarimetrically [38], the α-amino nitrogen by fluorometric methods [39] and the K and Na by photoelectric flame photometry [38] were measured. The molasses content was defined in mval per 1000 g of pulp. White sugar yield (technological yield) was calculated from the formula proposed by Buchholz et al. [40]. Generally, the following traits were observed: plant density (PD), sugar beet yield (SBY), biological sugar yield (BSY), pure sugar yield (PSY), sugar content (SC), refined sugar content (RSC), the yield of preferential sugar (YPS), recoverable sugar (RS), potassium molasses (PM), sodium molasses (SM), α-amino nitrogen (α-AN), alkalinity factor (AF) and standard molasses losses (SML).

### 2.3. Statistical Analysis

The Shapiro-Wilk normality test [41] was used for verifying normality of the distribution of all 13 observed traits. A two-way (year, plant protection product) MANOVA test was performed. Two-way analyses of variance (ANOVA) were carried out. Differences between years and plant protection products were compared by Fisher's least significant differences (LSDs). The correlations between all pairs of observed traits were estimated using Pearson's linear correlation coefficients. Correlation coefficients were tested and presented in heatmaps. A canonical variance analysis (CVA) and Mahalanobis distances [42,43] were used for the multivariate comparison of combinations of years and plant protection products. The data were analyzed using GenStat v. 18.2 software (VSN International; Hemel Hempstead, UK).

## 3. Results

The years (Wilk's $\lambda$ = 0.013; $F_{26;84}$ = 25.07; $p < 0.001$) were statistically significantly different for all 13 traits. The plant protection products (Wilk's $\lambda$ = 0.222; $F_{66;202}$ = 25.07; $p = 0.210$) and years $\times$ plant protection products interaction (Wilk's $\lambda$ = 0.060; $F_{130;352}$ = 25.07; $p = 0.177$) were not significantly different for all 13 traits. ANOVA indicated that effects of year were significant for all the traits of the study, except $\alpha$-AN and AF (Table 2). The main effects of the plant protection product were significant for SBY, BSY, PSY and SM, but year $\times$ plant protection product interaction was significant for PD only (Table 2).

**Table 2.** *F*-statistics from two-way analysis of variance (ANOVA) of observed traits.

| Source of Variation | Year | Plant Protection Product | Year × Plant Protection Product Interaction |
|---|---|---|---|
| The number of degrees of freedom | 2 | 5 | 10 |
| Plant density, PD | 43.47 *** | 1.87 | 2.52 * |
| Sugar beet yield, SBY | 13.39 *** | 4.04 ** | 1.22 |
| Biological sugar yield, BSY | 32.5 *** | 3.12 * | 1.16 |
| Pure sugar yield, PSY | 37.27 *** | 3.14 * | 1.21 |
| Sugar content, SC | 338.64 *** | 1.95 | 1.3 |
| Refined of sugar content, RSC | 349.71 *** | 1.91 | 1.85 |
| The yield of preferential sugar, YPS | 77.79 *** | 0.86 | 1.63 |
| Recoverable sugar, RS | 5.57 ** | 0.47 | 0.9 |
| Potassium molasses, PM | 5.15 ** | 0.77 | 0.59 |
| Sodium molasses, SM | 85.36 *** | 3.17 * | 1.77 |
| $\alpha$-amino nitrogen, $\alpha$-AN | 2.68 | 0.49 | 0.64 |
| Alkalinity factor, AF | 0.42 | 0.28 | 0.2 |
| Standard molasses losses, SML | 6.37 ** | 0.67 | 0.81 |

\* $p < 0.05$; ** $p < 0.01$; *** $p < 0.001$.

Mean values and standard deviations for the all traits indicated a high variability among the tested years, for which significant differences were found in terms of 11 out of 13 analyzed quantitative traits (Figures 1–13).

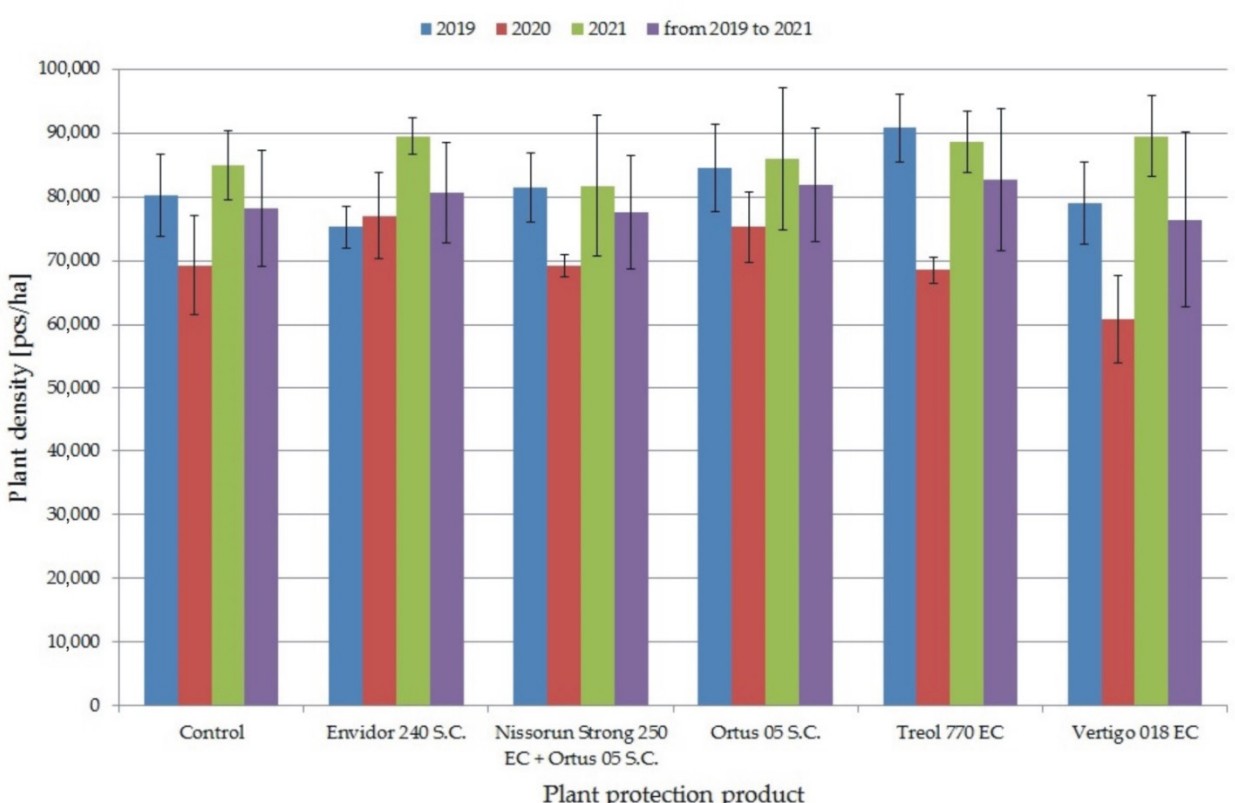

**Figure 1.** Mean values for plant density over the years.

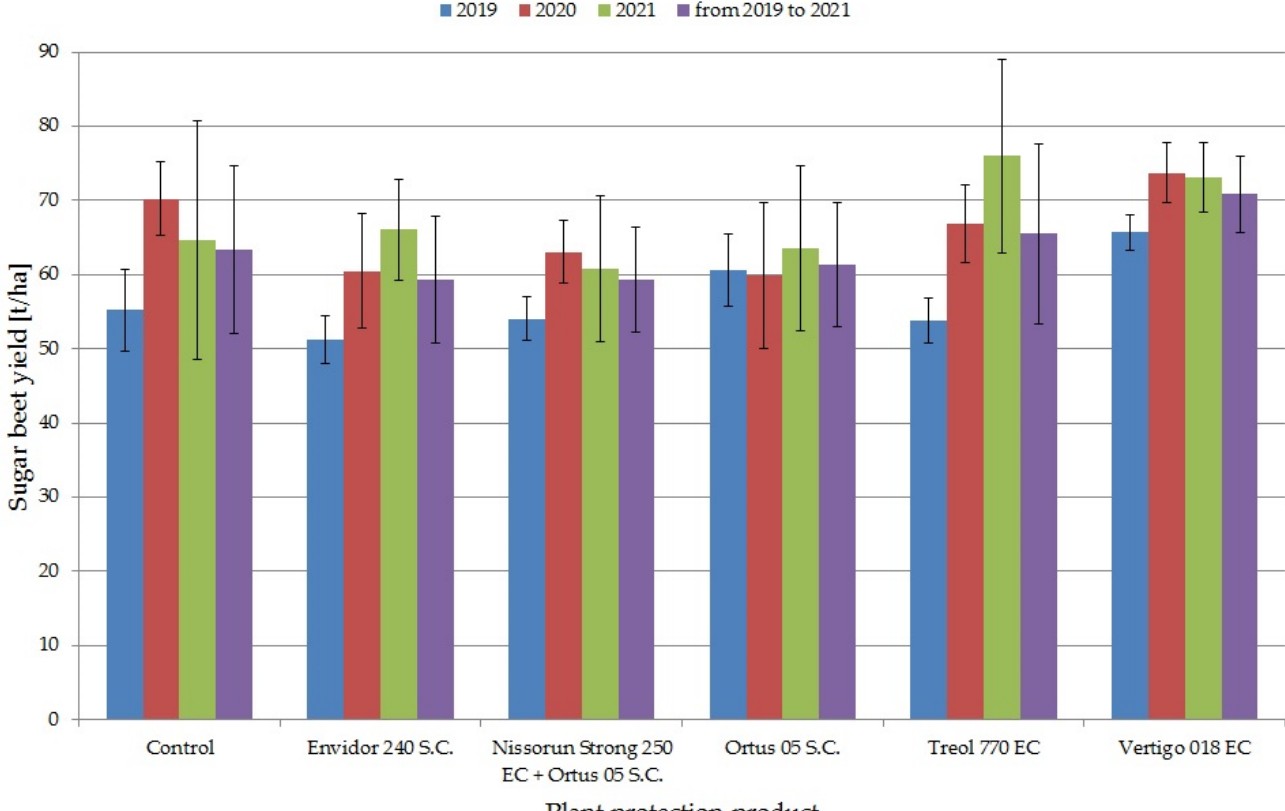

**Figure 2.** Mean values for sugar beet yield over the years.

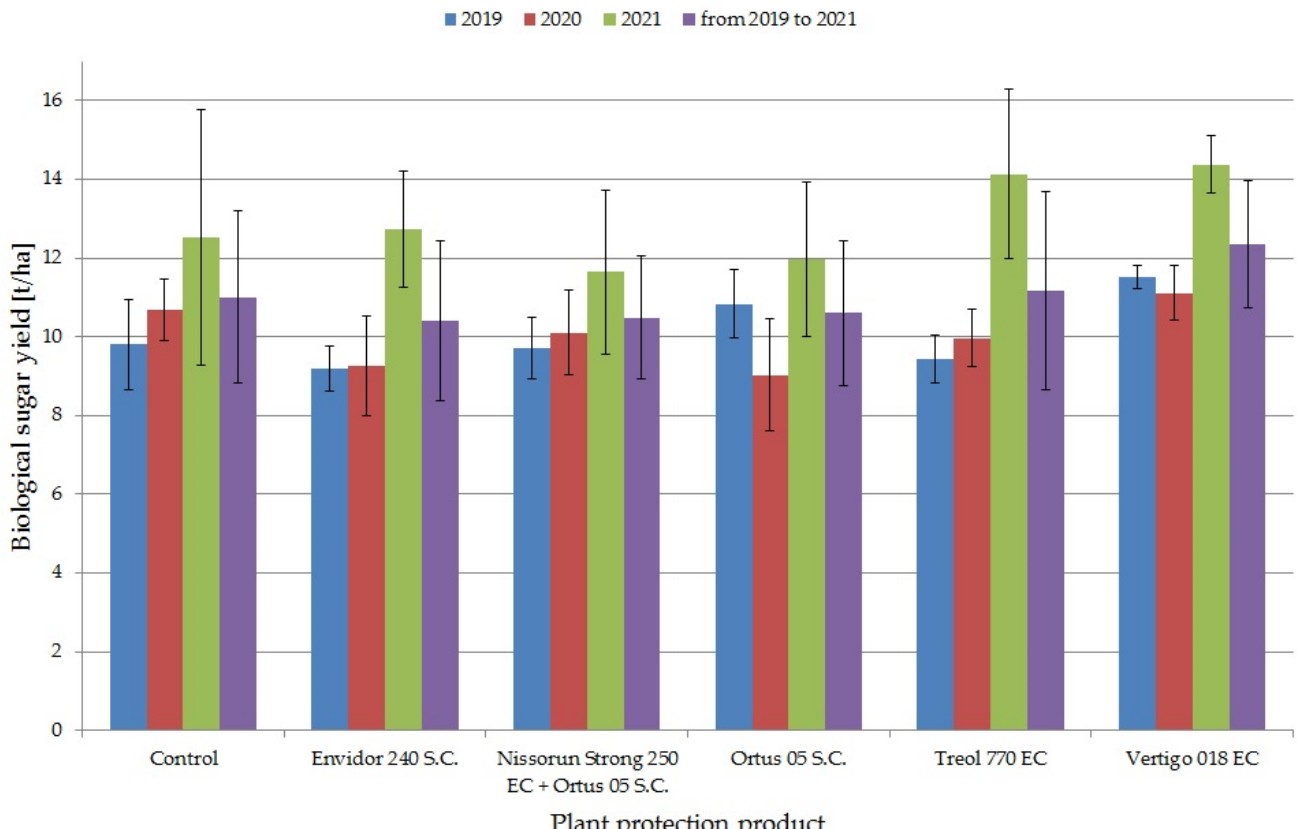

**Figure 3.** Mean values for biological sugar yield over the years.

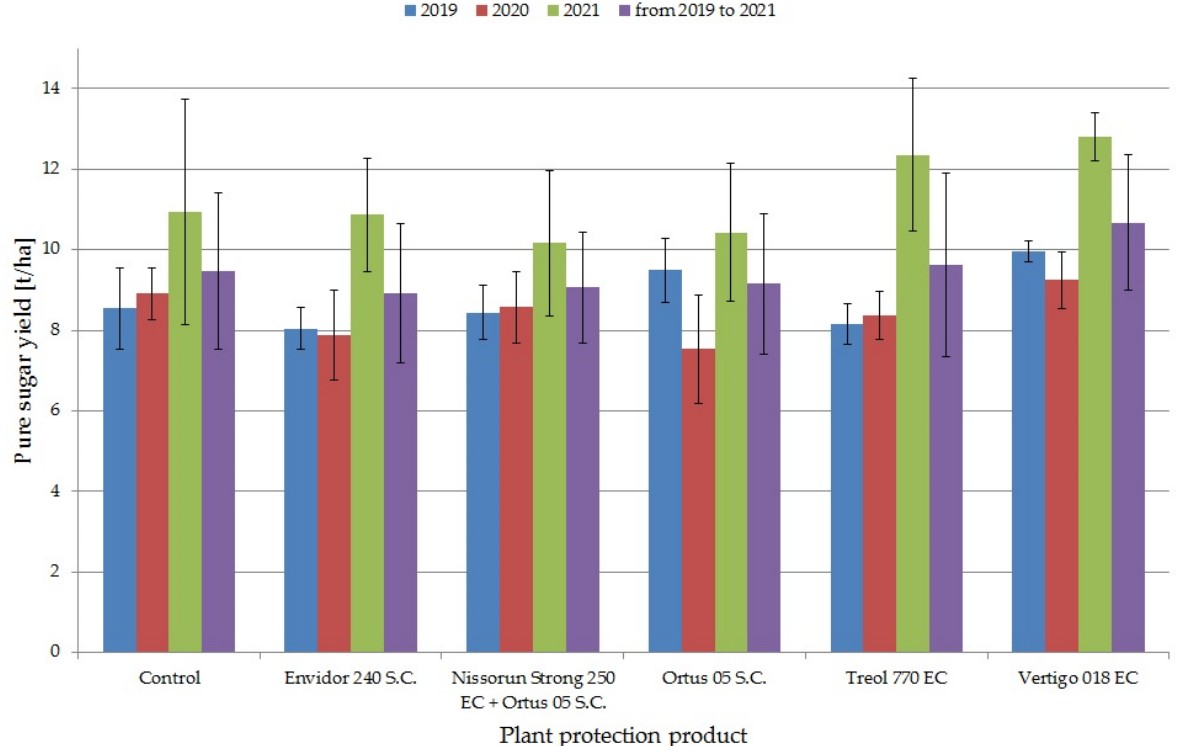

**Figure 4.** Mean values for pure sugar yield over the years.

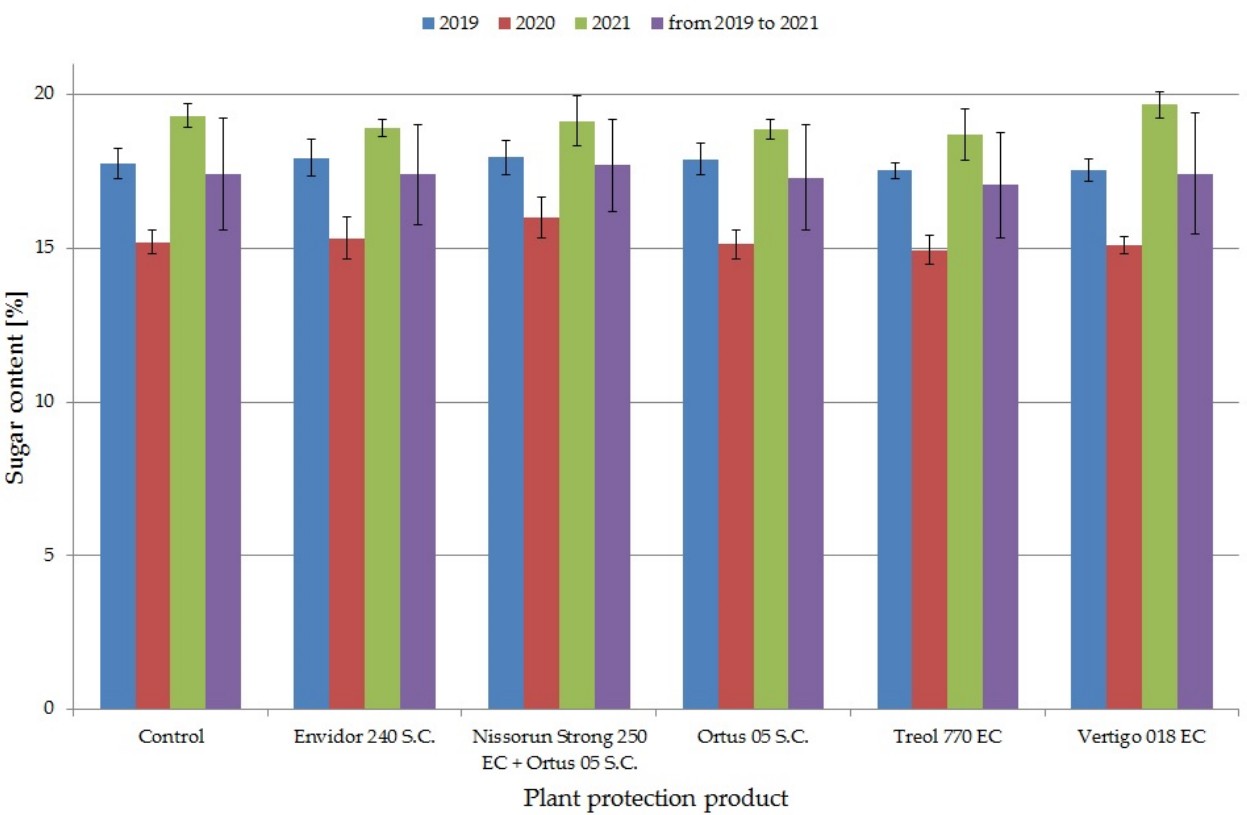

**Figure 5.** Mean values for sugar content over the years.

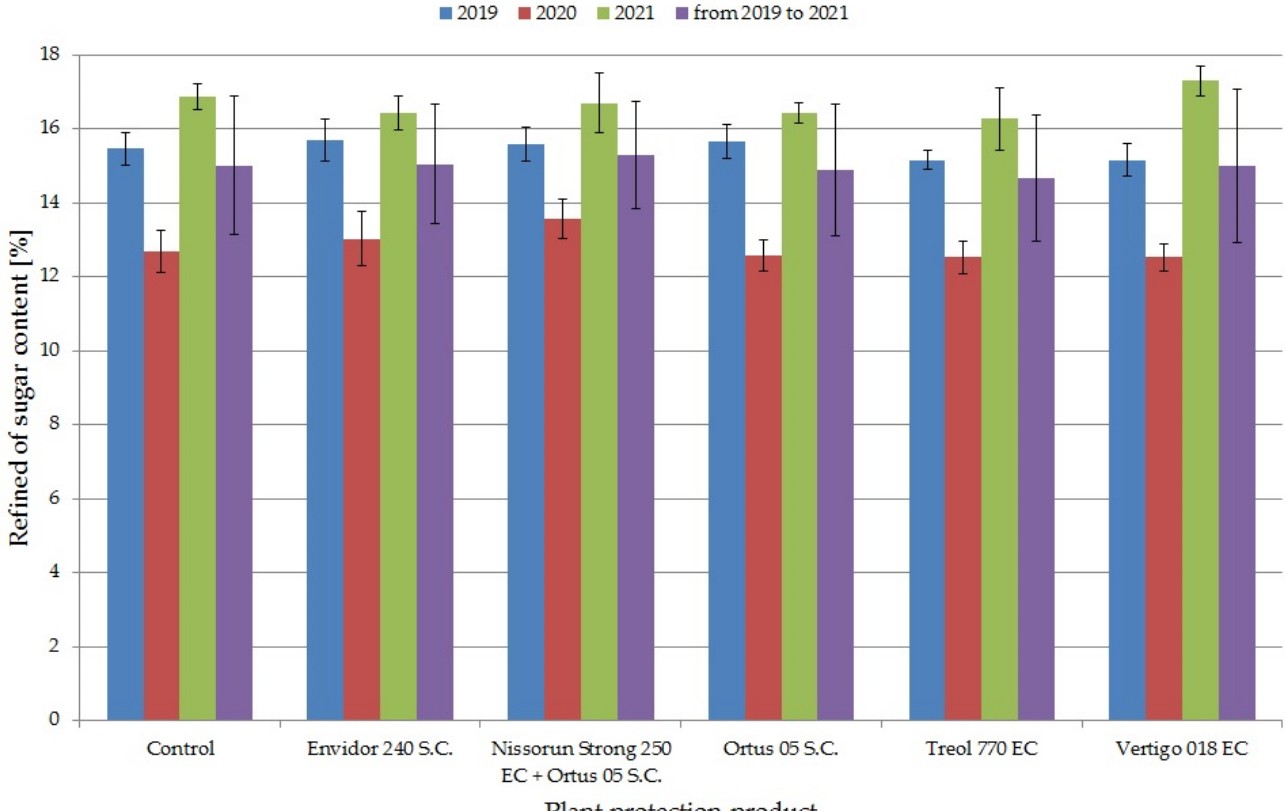

**Figure 6.** Mean values for refined of sugar content over the years.

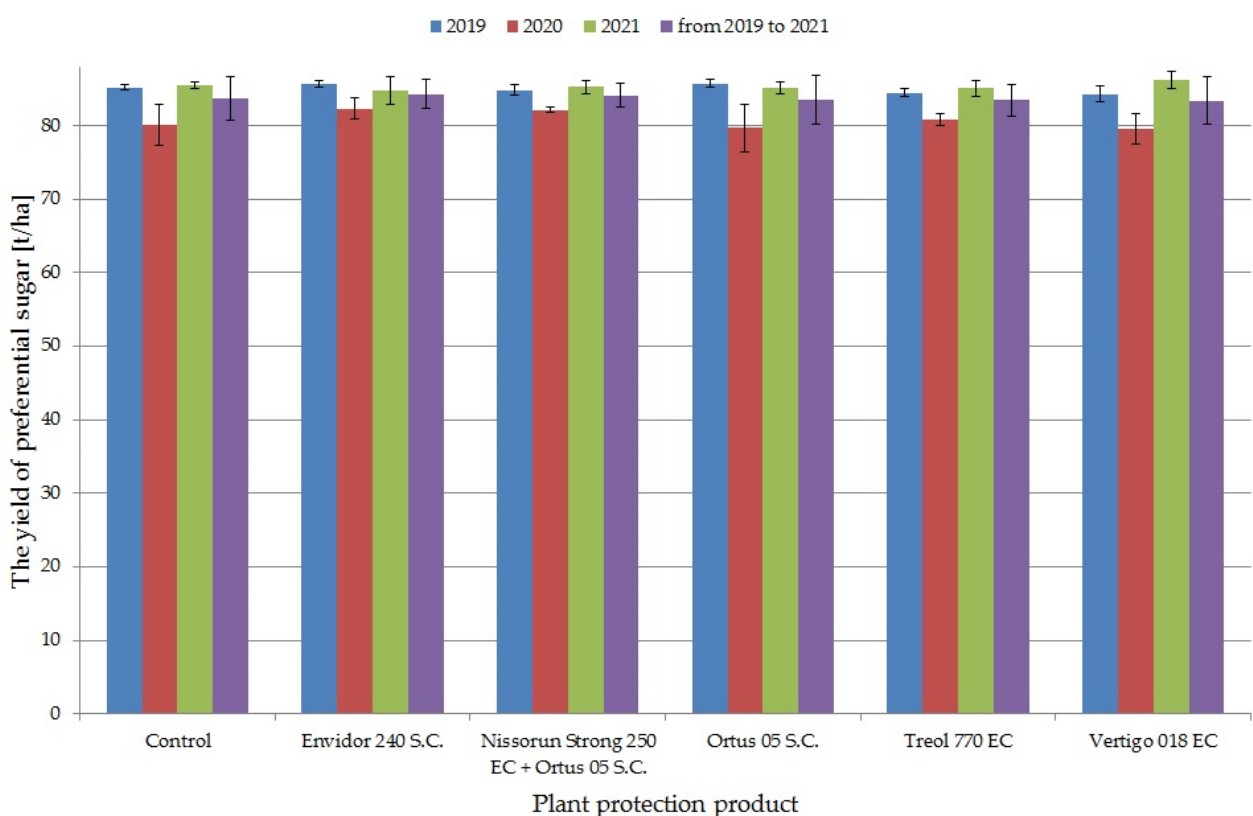

**Figure 7.** Mean values for the yield of preferential sugar over the years.

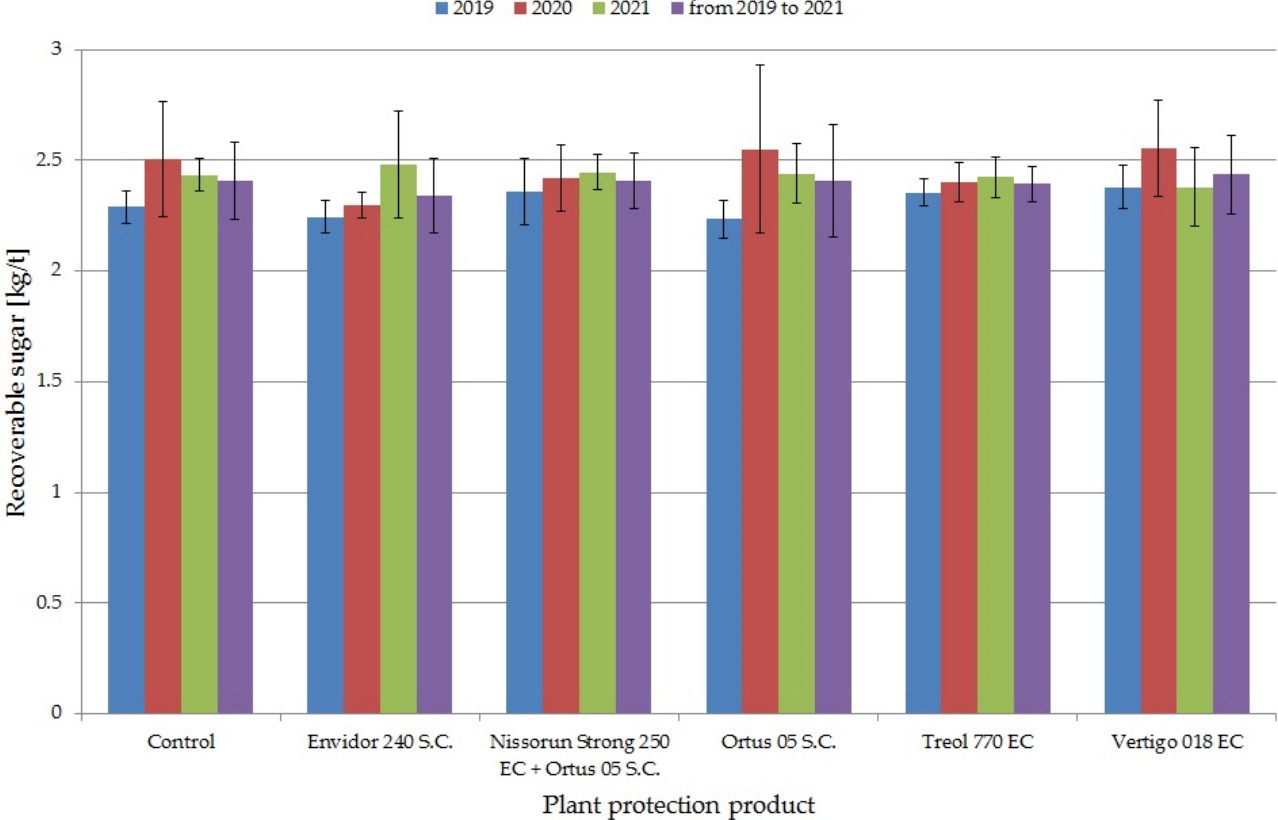

**Figure 8.** Mean values for recoverable sugar over the years.

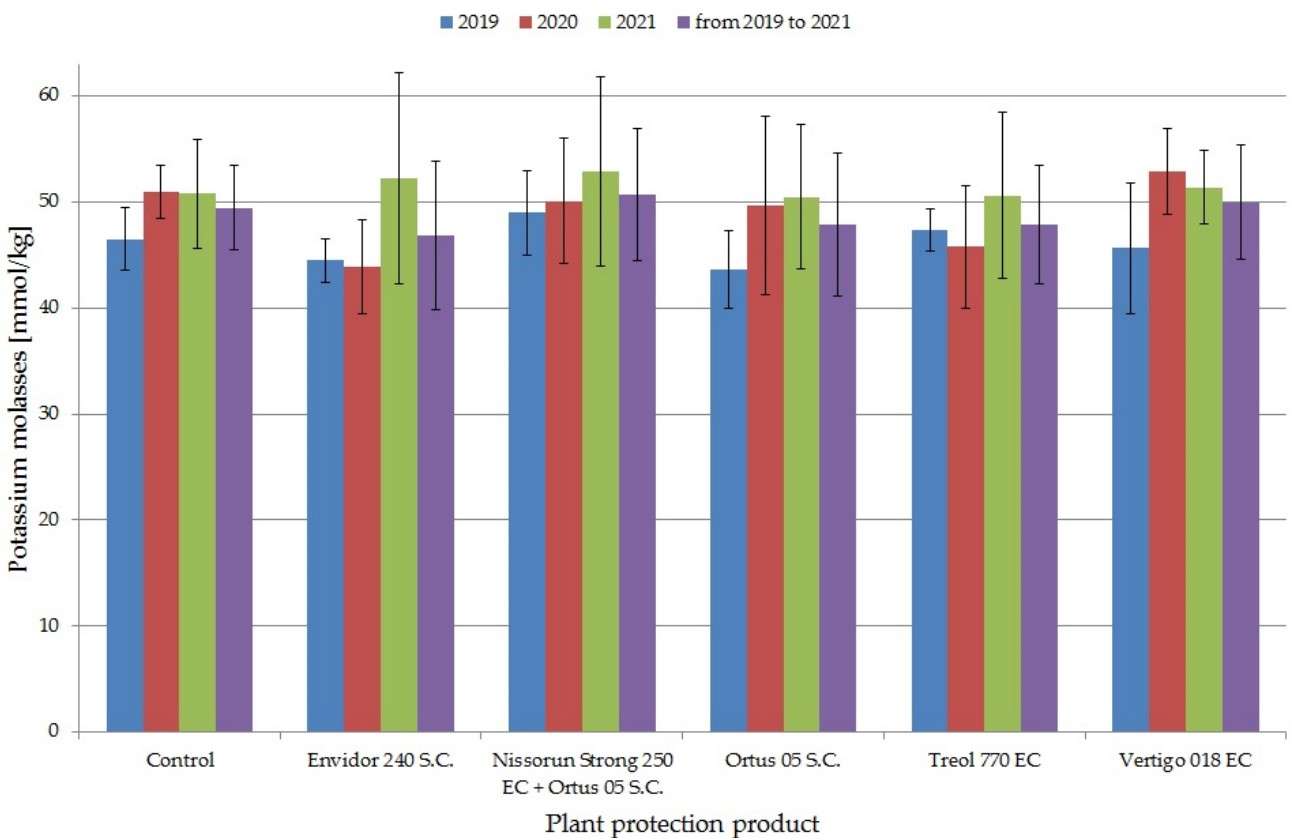

**Figure 9.** Mean values for potassium molasses over the years.

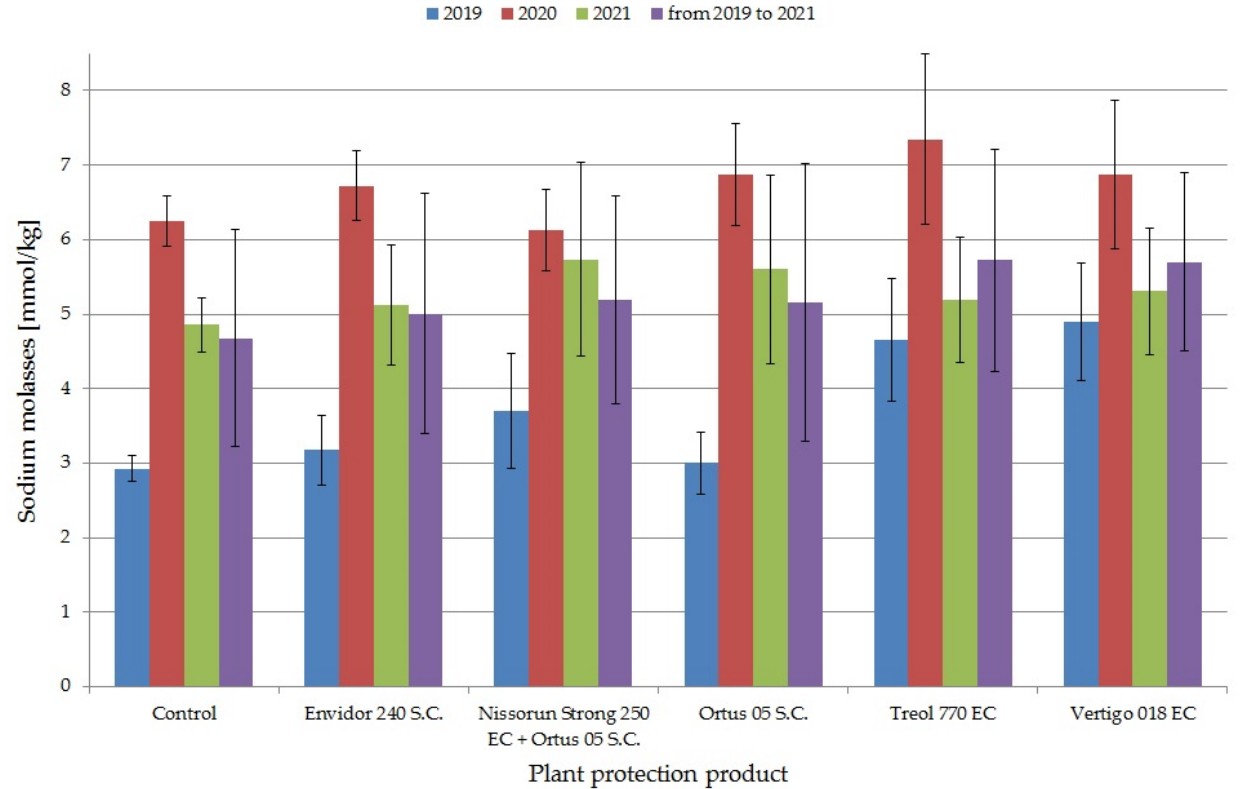

**Figure 10.** Mean values for sodium molasses over the years.

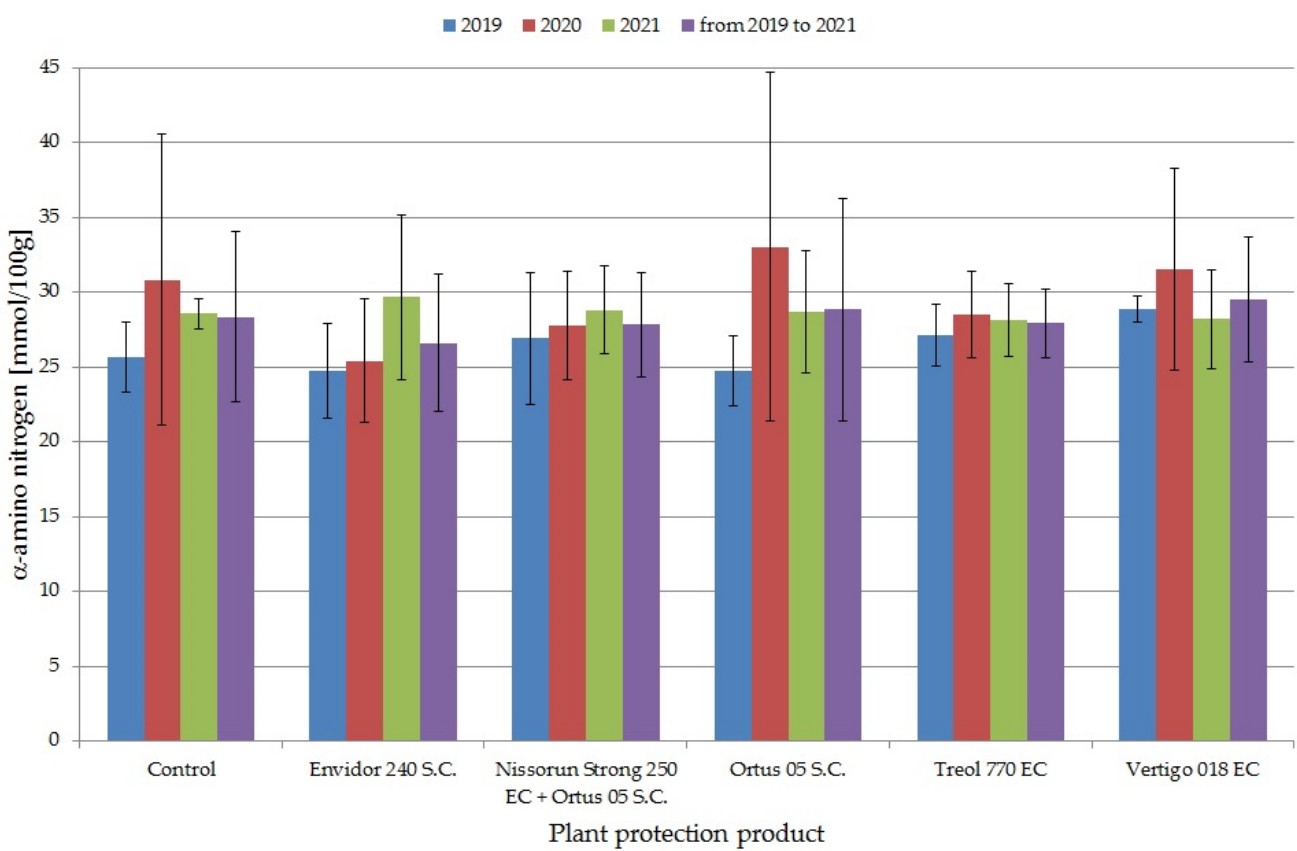

**Figure 11.** Mean values for α-amino nitrogen over the years.

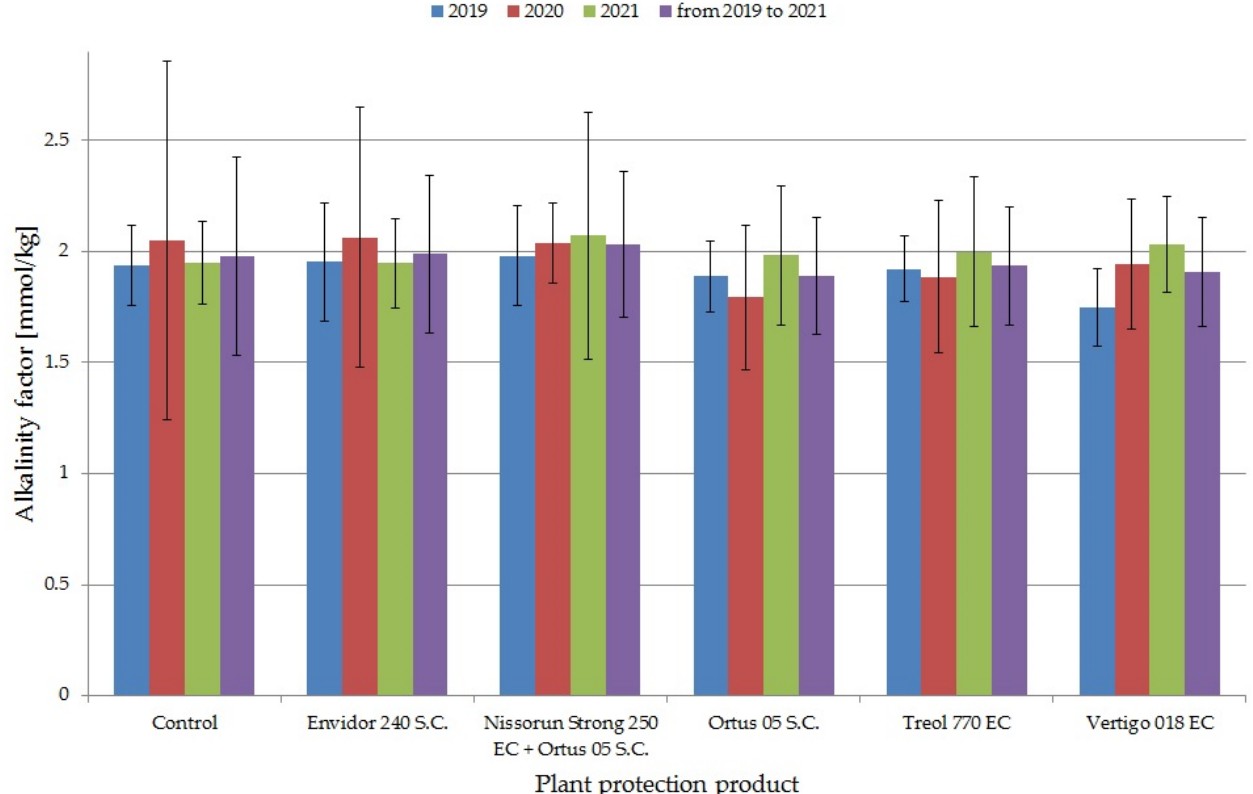

**Figure 12.** Mean values for alkalinity factor over the years.

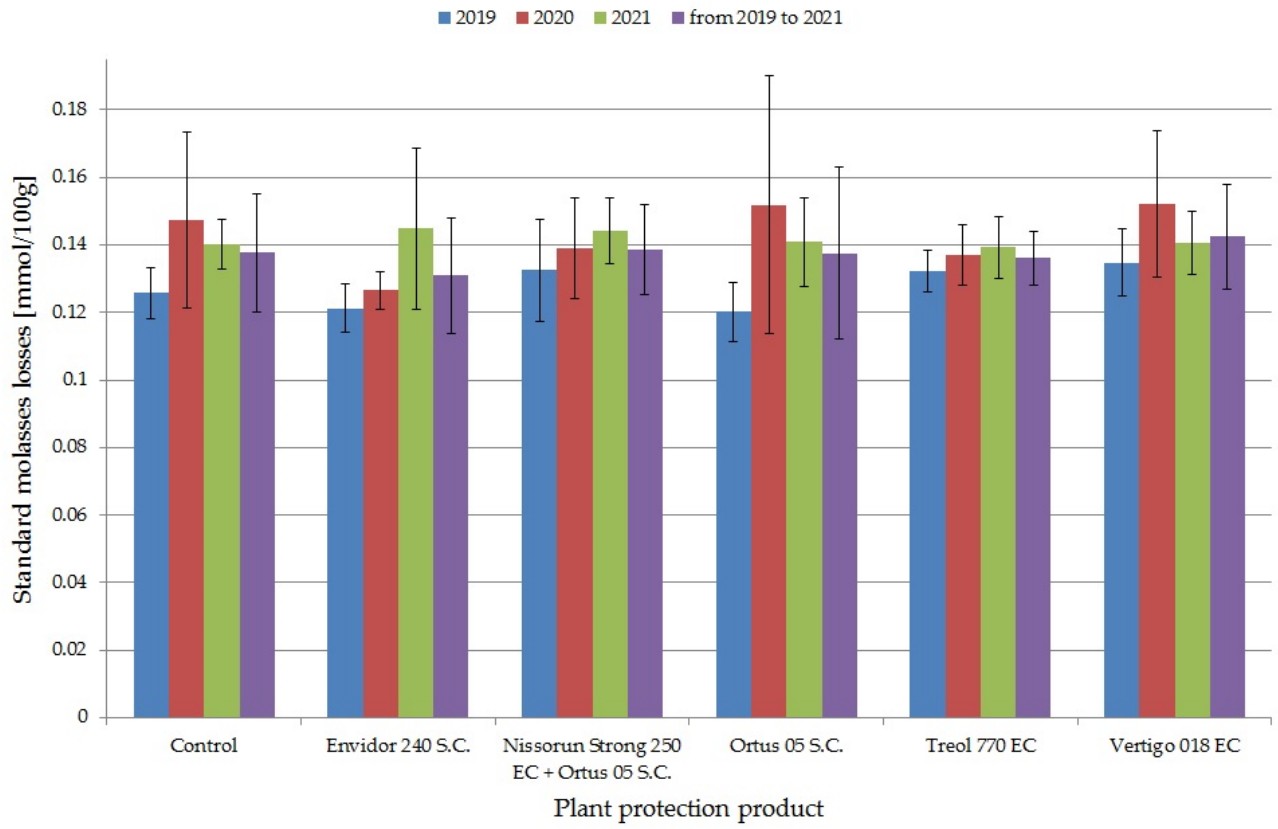

**Figure 13.** Mean values for standard molasses losses over the years.

Positive correlations were observed for the following pairs of traits: PD-BSY, PD-PSY, PD-SC, PD-RSC, PD-YPS, SBY-BSY, SBY-PSY, SBY-SM, BSY-PSY, BSY-SC, BSY-RSC, BSY-YPS, PSY-SC, PSY-RSC, PSY-YPS, SC-RSC, SC-YPS, RSC-YPS, RS-PM, RS-SM, RS-$\alpha$-AN, RS-SML, PM-$\alpha$-AN, PM-AF, PM-SML, SM-$\alpha$-AN, SM-SML and $\alpha$-AN-SML (Figure 14, Table 3). Negative statistically significant relationships were observed between the following pairs of traits: SM-PD, SM-SC, SM-RSC, YPS-RS, YPS-PM, YPS-SM, YPS-$\alpha$-AN, YPS-SML, AF-RS, AF-$\alpha$-AN and AF-SML (Figure 14, Table 3).

**Table 3.** The correlation matrix for 13 observed traits.

| Trait | PD | SBY | BSY | PSY | SC | RSC | YPS | RS | PM | SM | $\alpha$-AN | AF |
|---|---|---|---|---|---|---|---|---|---|---|---|---|
| SBY | −0.02 | | | | | | | | | | | |
| BSY | 0.38 ** | 0.83 *** | | | | | | | | | | |
| PSY | 0.42 *** | 0.78 *** | 0.99 *** | | | | | | | | | |
| SC | 0.65 *** | −0.03 | 0.53 *** | 0.59 *** | | | | | | | | |
| RSC | 0.65 *** | −0.05 | 0.51 *** | 0.59 *** | 1.00 *** | | | | | | | |
| YPS | 0.56 *** | −0.13 | 0.35 * | 0.44 *** | 0.82 *** | 0.87 *** | | | | | | |
| RS | −0.11 | 0.15 | 0.06 | 0.01 | −0.1 | −0.2 | −0.65 *** | | | | | |
| PM | 0.07 | 0.18 | 0.21 | 0.17 | 0.13 | 0.05 | −0.32 ** | 0.75 *** | | | | |
| SM | −0.36 ** | 0.40 *** | 0.06 | 0 | −0.52 *** | −0.55 *** | −0.63 *** | 0.38 ** | 0.2 | | | |
| $\alpha$-AN | −0.13 | 0.05 | −0.01 | −0.06 | −0.1 | −0.18 | −0.61 *** | 0.91 *** | 0.46 *** | 0.30 * | | |
| AF | 0.09 | 0.13 | 0.13 | 0.13 | 0.02 | 0.06 | 0.23 | −0.34 ** | 0.26 * | 0.08 | −0.67 *** | |
| SML | −0.1 | 0.15 | 0.09 | 0.03 | −0.07 | −0.17 | −0.62 *** | 0.98 *** | 0.76 *** | 0.40 *** | 0.92 *** | −0.35 ** |

* $p < 0.05$; ** $p < 0.01$; *** $p < 0.001$; PD—plant density, SBY—sugar beet yield, BSY—biological sugar yield, PSY—pure sugar yield, SC—sugar content, RSC—refined of sugar content, YPS—the yield of preferential sugar, RS—recoverable sugar, PM—potassium molasses, SM—sodium molasses, $\alpha$-AN—$\alpha$-amino nitrogen, AF—alkalinity factor, SML—standard molasses losses.

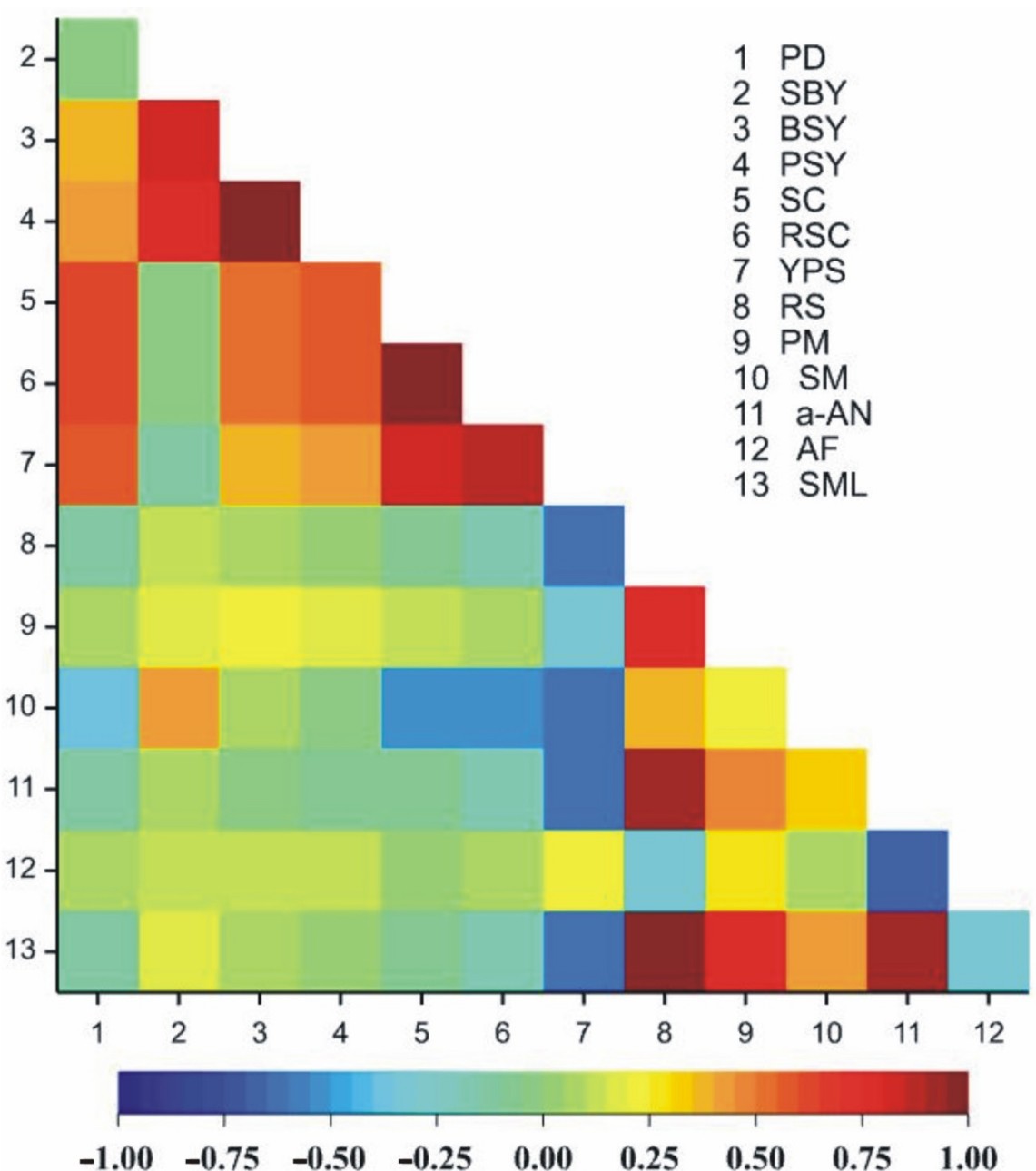

**Figure 14.** Heatmaps for correlation coefficients between all pairs of the 13 observed traits ($r_{cr}$ = 0.23). PD—plant density, SBY—sugar beet yield, BSY—biological sugar yield, PSY—pure sugar yield, SC—sugar content, RSC—refined of sugar content, YPS—the yield of preferential sugar, RS—recoverable sugar, PM—potassium molasses, SM—sodium molasses, $\alpha$-AN—$\alpha$-amino nitrogen, AF—alkalinity factor, SML—standard molasses losses.

Analysis of the first two canonical variates for 18 combinations regarding the 13 quantitative traits is shown in Figure 15. The first two canonical variates accounted for 85.49% of the total variability between the individual combinations (Figure 15). The significant positive relationship with the first canonical variate was found for PD, BSY, PSY, SC, RSC, YPS, but negative for SM. The $CV_2$ was negatively correlated with: SBY, BSY, PSY, RS, PM, SM, $\alpha$-AN and SML. The greatest variation in terms of all the 13 traits jointly was found for V in 2020 and V in 2021 (Mahalanobis distance between them amounted to 13.08). The greatest similarity was found between C in 2019 and O in 2019 (1.26) (Table 4).

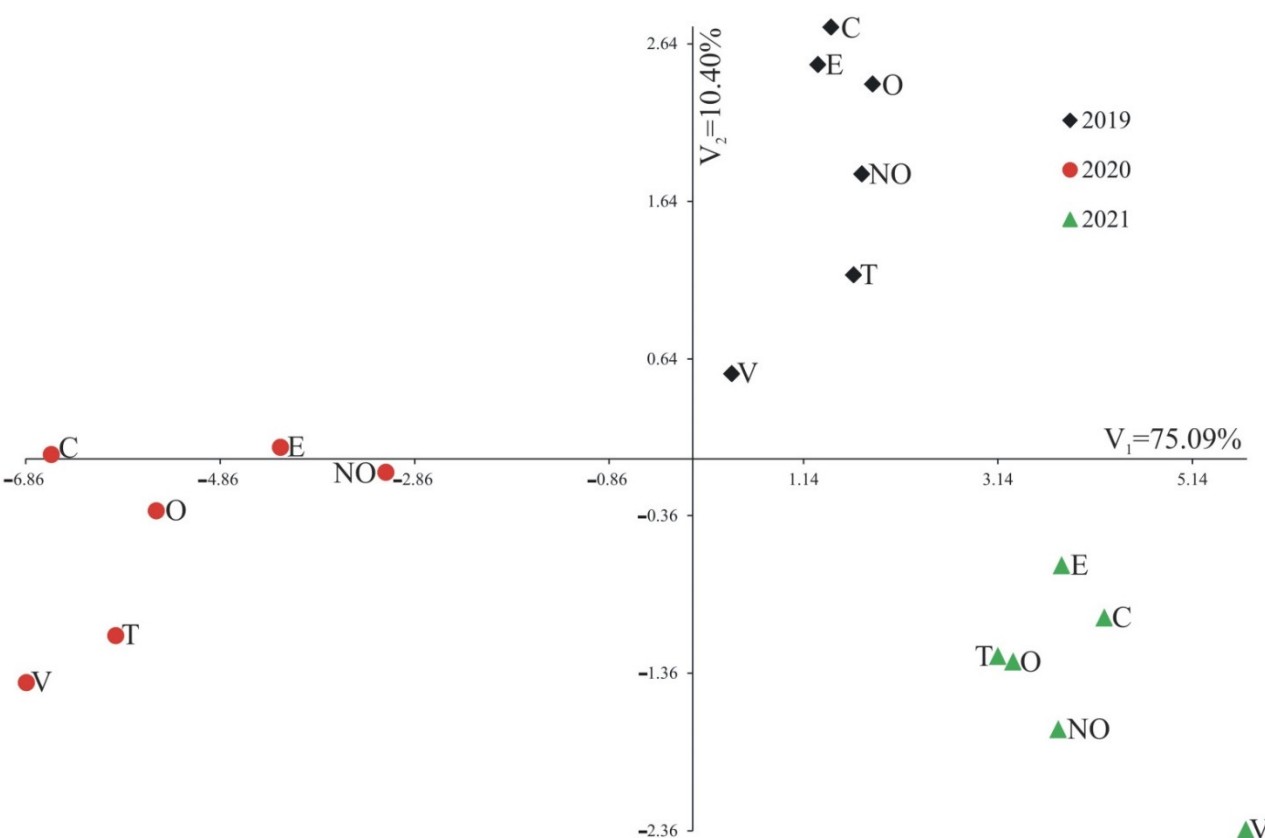

**Figure 15.** Distribution of 18 combinations of years and plant protection products in the space of the first and second canonical variables.

**Table 4.** Mahalanobis distances between combinations of plant protection products and years.

| Year | | | | 2019 | | | | | 2020 | | | | | 2021 | | | |
| | Plant Protection Product | NO | C | V | E | O | T | NO | C | V | E | O | T | NO | C | V | E | O |
|---|---|---|---|---|---|---|---|---|---|---|---|---|---|---|---|---|---|---|
| | C | 1.43 | | | | | | | | | | | | | | | | |
| | V | 2.62 | 3.20 | | | | | | | | | | | | | | | |
| 2019 | E | 1.67 | 1.31 | 3.41 | | | | | | | | | | | | | | |
| | O | 1.79 | 1.26 | 3.13 | 1.99 | | | | | | | | | | | | | |
| | T | 2.42 | 3.45 | 2.93 | 3.76 | 3.28 | | | | | | | | | | | | |
| | NO | 5.55 | 5.74 | 4.24 | 5.68 | 6.23 | 5.77 | | | | | | | | | | | |
| | C | 8.88 | 8.77 | 7.60 | 8.68 | 9.01 | 9.25 | 5.24 | | | | | | | | | | |
| 2020 | V | 9.41 | 9.43 | 8.00 | 9.22 | 9.70 | 9.79 | 5.02 | 2.70 | | | | | | | | | |
| | E | 6.66 | 6.85 | 5.37 | 6.77 | 7.12 | 6.24 | 2.56 | 4.80 | 5.16 | | | | | | | | |
| | O | 7.83 | 7.89 | 6.65 | 7.82 | 8.22 | 7.71 | 3.61 | 3.57 | 4.04 | 2.44 | | | | | | | |
| | T | 8.36 | 8.53 | 6.81 | 8.35 | 8.77 | 8.16 | 3.50 | 3.79 | 3.05 | 2.63 | 2.46 | | | | | | |
| | NO | 4.23 | 5.40 | 4.71 | 5.14 | 5.16 | 4.43 | 7.49 | 10.89 | 10.88 | 8.65 | 9.75 | 9.90 | | | | | |
| | C | 3.90 | 4.87 | 4.60 | 4.75 | 4.45 | 4.36 | 7.81 | 11.15 | 11.23 | 8.98 | 10.05 | 10.30 | 1.68 | | | | |
| 2021 | V | 6.53 | 7.05 | 7.17 | 7.16 | 6.58 | 6.93 | 9.77 | 13.02 | 13.08 | 10.84 | 11.73 | 12.19 | 4.33 | 3.97 | | | |
| | E | 3.97 | 4.94 | 4.27 | 5.09 | 4.65 | 4.07 | 7.50 | 10.91 | 11.14 | 8.66 | 9.77 | 10.09 | 2.91 | 2.35 | 5.28 | | |
| | O | 3.80 | 5.04 | 3.90 | 4.91 | 4.63 | 3.51 | 7.09 | 10.41 | 10.53 | 8.05 | 9.27 | 9.38 | 1.60 | 1.76 | 5.07 | 2.52 | |
| | T | 3.85 | 4.64 | 3.64 | 4.92 | 4.03 | 4.03 | 6.91 | 10.08 | 10.28 | 8.01 | 9.10 | 9.33 | 2.92 | 2.23 | 4.17 | 2.78 | 2.33 |
| | | | | | | | | | | | | Dα = 9.73 | | | | | | |

## 4. Discussion

The "pest's food" spectrum is extremely wide. It includes over 300 species belonging to various systematic groups, growing wild and cultivated, for example, under cover, from where they can move to the surrounding fields. In addition, the spider mite spends several generations per season and has an exceptionally high reproductive potential, which means that the development of the generation can even lead to a 100-fold increase in the population size!

Apart from beetroot the spider mite also feeds intensively on sugar beet, fodder and sugar maize varieties and their colonization is favored by the "stay green" feature, thanks to which the plants remain green for longer.

Generally application of acaricides is adopted as a quick method to destroy the maximum number of mites and stop their multiplication. The strategic treatment schedule is applied once or twice per vegetation season (field plantations) to stop their development and multiplication. Mite control with chemical acaricides was a very popular method at one time and was partially successful, but the drawbacks of using synthetic acaricides include harmful residual effect on natural environment and can cause contamination of food, especially fruit and vegetables meant to be consumed when freshly harvested.

Continuous use of acaricides also results in the development of resistance of mite strains. Mites are resistant to the most commonly used acaricides including organophosphates, pyrethroids, carbazinates, quinolines, carbamates, tetrazines, diphenyl oxazolines, quinazolines, phenoxy-pyrazoles, thiazolidines, macrocyclic lactones, pyridazones and pyrazole derivatives.

In Poland, the assortments for use in beet are very limited. Only one preparation is registered—Ortus 05 SC, which fights only spider mite, not eggs. It works by contact and in order to be fully effective it must come into direct contact with the pest's body, which in practice is not that simple, and not only because of the "protective spider web". Beet plants, although sown in wide row spacing, at some point begin to form a thicket. The leaves overlap and a compact "umbrella" is formed, which makes it difficult to penetrate the field and thoroughly cover the bottom side of the beet leaves with the working liquid, which is the only condition for obtaining satisfactory results. Therefore, the effectiveness of the control depends largely on the technical side of the treatment; in accordance with the recommendations of the acaricide manufacturer, for it to work effectively, both the dose of water ($400 \, l \, ha^{-1}$) and the preparation ($1.5$–$1.8 \, l \, ha^{-1}$) should not be reduced. These methods are compatible with the EU strategy and the IPM directive. In our research, we used preparations that are registered for the control of spider mites in orchards, ornamental plants and do not have label extension for field plants, specifically sugar beet.

Recently, many studies have been carried out on replacing synthetic acaricides with new, safer agents, due to the risk of developing tolerance, toxicity and harmfulness to the natural environment associated with their overuse.

The pure sugar yield is a final product in sugar beet production. The largest increases in pure sugar yield were obtained in 2021. In this year plants were exposed to drought stress in May. Smaller increases were observed in 2019, when the value of the soil moisture was high and there was no drought stress. This confirms the results of previous studies, which prove the effect of field experiments of other authors [44]. Pure sugar yield is determined by the biological yield of sugar, which depends on the yield of roots, sugar content and the content of molasses-forming components. Root yield has the greatest influence on pure sugar yield [44,45].

The control of *T. urticae* relies mainly on the use of synthetic acaricides. This is not always effective, as this species has a high ability to develop resistant populations [46–48]. According to regulations on integrated pest management, plants should be sprayed when threshold values of economic injury are exceeded and when the pest population cannot be reduced by growing arachnid-tolerant plants or using biological methods. If several treatments are necessary, they should be performed with products from different chemical groups, representing different modes of action. Currently, selected products for the control of spider mites with different crops can be chosen from among 14 chemical groups (15 active substances) available on the world market of plant protection chemicals. There are insecticides, insecto-acaricides and acaricides. The presented pesticides are diverse in their mechanism of action. They affect the nervous system—neurotoxins (5)—inhibit the growth of mites or disturb their development (8) and inhibit lipid metabolism (2) [46,49].

In Poland, only one pesticide, Ortus 05 SC, is registered in sugar beet for the control of spider mites. Fenpyroximate (a compound from the phenoxy-pyrazole group) fights

only mobile forms of mites, not eggs. Most populations of spider mites developed resistance to chemical groups within insecticides after a few years of use. Resistance to the ovicide clofentezine developed quite rapidly and cross-resistance to hexythiazox also occurred [50]. Al-Jboory et al. [51] reported that a bromo-propylate-resistant strain (R) of *T. urticae* showed strong positive cross-resistance towards insecticide, insecto-acaricide and acaricide, moderate positive cross-resistance towards amitraz and low negative cross-resistance towards chlorpyrifos. Resistance was detected in a relatively short period of time, leading to decreased susceptibility to all the compounds in this group [52].

In our research, we used chemical preparations and acaricide that are often used for treatments in orchards, as well as in vegetables and strawberries. The efficacy of abamectin and paraffin oil was observed. In experiments, products based on these active substances showed greater efficiency on the biological and technological yield of sugar beets. Environmental condition effects on biological and commercial traits of sugar beet is very significant. The field trials demonstrated the impact of weather conditions, especially the influence of temperature and precipitation on development of plants also development stadium of mites on the sugar beet yield. The obtained results show that weather conditions in different years during the growing season of sugar beet have a huge impact on the root yield, sugar content and the formation of assimilates in the plant growth and development process. A greater number of organic compounds produced at the beginning of vegetation is used for leaf growth, which in turn reduces the initial share of roots in the total plant mass [53]. Our field observations show that at this time sugar beet plants may be very sensitive to mite feeding. In the years of research, it was found that spider mites were present in mass quantities in the full growing season, i.e., in July and August. During this period, the plant begins to accumulate large amounts of sugar in the roots. Plants infested with *Tetranychus urticae* defend themselves by creating a rosette of young leaves at the expense of sugar assimilation in the roots. This can result in a reduced polarization in the roots. In the years 2019–2020, a lower polarization than in the year of completion of the field experiment was observed. In our research, the polarization was the highest in the last year of the research. This may have been influenced by application of plant protection products against *Tetranychus urticae*.

## 5. Conclusions

Plant protection products used for protection affect the content of sugar beet yield, biological sugar yield, pure sugar yield and sodium molasses. Vertigo 018 EC, Treol 770 EC and Ortus 05 SC were effective on *Tetranychus urticae* population. The best results of SBY, BSY and SY were observed after application of the (V) variant. The technological yield of sugar beet was determined directly proportionally by the mass of roots and sugar content with the use of any decision-making method different acaricides. The technological yield of sugar increases with the length of the growing season, i.e., the number of days between sowing and harvesting. The sodium content did not have a statistically significant effect on the technological yield of sugar beet for any of the applied treatments for two-spotted spider mites.

**Author Contributions:** Conceptualization, J.B. and M.J.; methodology, J.B. and M.J.; software, J.B.; validation, J.B.; formal analysis, J.B.; investigation, M.J.; resources, J.B.; data curation, M.J.; writing—original draft preparation, J.B., M.J., D.Z. and R.D.; writing—review and editing, J.B., M.J. and R.D.; visualization, J.B.; supervision, J.B. and M.J.; project administration, M.J.; funding acquisition, M.J. All authors have read and agreed to the published version of the manuscript.

**Funding:** This research was part of targeted subsidy in Ministry of Science and Higher education in Poland (2011–2023), implemented in Institute of Plant Protection–NIR.

**Institutional Review Board Statement:** Not applicable.

**Informed Consent Statement:** Not applicable.

**Data Availability Statement:** Not applicable.

**Conflicts of Interest:** The authors declare no conflict of interest.

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
