# Peer review of "The Effect of Acaricide Control of the Two-Spotted Spider Mite Tetranychus urticae Koch on the Cultivation of Sugar Beet (Beta vulgaris L.) and on the Size and Quality of the Yield"

_applsci, doi:10.3390/app122312139_

Round 1

Reviewer 1 Report

I suggest you modify the title:  Field Insecticidal control of Tetranychus urticae Koch in Sugar Beet (Beta vulgaris L)

 Abstract: You must show here some of your results.

 Keywords: Don’t use words in the title.

 Introduction: Well documented but there are several English issues.

 Materials and Methods: Rewrite please. The first paragraph must be on the description of the experimental location and its soil characteristics.

 Table 1: Pesticide free instead of No pesticide

 Data collection: need to be rewritten clearly because it is still hard to understand.

 Statistical analysis: Rewrite please

 Table 3 - 15: I suggest you use here histograms with s.d. top bars along the years.

 Conclusion: I suggest you write it in only one paragraph.

 In general, there is serious English issues. So, a requirement of natives or editing service would be helpful.

Author Response

Response to Reviewer 1 Comments

Reviewer #1

Point 1: I suggest you modify the title:  Field Insecticidal control of Tetranychus urticae Koch in Sugar Beet (Beta vulgaris L).

Response: We modified the title to: “The Effect of Acaricide Control on the Two Spotted Spider Mite Tetranychus urticae Koch on the Cultivation of Sugar Beet (Beta vulgaris L.) and on the Size and Quality of the Yield”.

Point 2: Abstract: You must show here some of your results.

Response: We added our results in abstract: “). The years were statistically significantly different for all 13 traits.” and “Positive correlations were observed for 28 pairs of traits, however negative statistically significant relationships were observed between 11 pairs of traits. The first two canonical variates accounted for 85.49% of the total variability between the individual combinations. The significant positive relationship with the first canonical variate was found for PD, BSY, PSY, SC, RSC, YPS, while the negative – for SM. The CV2 was negatively correlated with: SBY, BSY, PSY, RS, PM, SM, a-AN and SML. The greatest variation in terms of all the 13 traits jointly was found for Vertigo 018 EC in 2020 and Vertigo 018 EC in 2021. The greatest similarity was found between control in 2019 and Ortus 05 SC in 2019.”.

Point 3: Keywords: Don’t use words in the title.

Response: We corrected keywords. New keyword are: insecticidal protection; yield quality; efficiency; canonical variance analysis; Mahalanobis distances; Pearson's linear correlation.

Point 4: Introduction: Well documented but there are several English issues.

Response: Has been corrected by a Native speaker.

Point 5: Materials and Methods: Rewrite please. The first paragraph must be on the description of the experimental location and its soil characteristics.

Response:  Has been corrected.

Point 6: Table 1: Pesticide free instead of No pesticide.

Response: We changed “No pesticide” to “Pesticide free”.

Point 7: Data collection: need to be rewritten clearly because it is still hard to understand.

Response: Has been corrected by a Native speaker.

Point 8: Statistical analysis: Rewrite please.

Response: We corrected description of “Statistical analysis” subsection.

Point 9: Table 3 - 15: I suggest you use here histograms with s.d. top bars along the years.

Response: We changed the tables to figures..

Point 10: Conclusion: I suggest you write it in only one paragraph.

Response: We rewrote Conclusion section as one paragraph.

Point 11: In general, there is serious English issues. So, a requirement of natives or editing service would be helpful.

Response: Has been corrected by a Native speaker

Reviewer 2 Report

Row

comments

1-3

TITLE: Due, that the goal is to control Tetranychus urtice and influence on cultivation of sugar beet the title should be: The Effect of Acaricide Control on the Two Spotted Spider Mite Tetranychus urticae Koch in the Cultivation of Sugar Beet Beta vulgaris L. and on the Size and Quality of the Yield

14

In the years 2019-2021, field experiments were carried out (WHERE?),

15

Before beet always add “sugar” where it is missing during the text. This is for better understanding< the
purpose of which was to test the insecticidal and acaricidal effectiveness of
(SUGAR?)
 beet cultivation protection against T. urticae (italic) and to assess its impact on the size and quality of the (SUGAR?) beet crop.

16

In the experiment used were acaricides:  spirodiclofen – 240 g 16 – 22.11%, mixture of hexythiazox – 250 g – 23.15% and fenpyroximate – 51.2 g – 5.02%, fenpyroximate 17 – 51.2 g – 5.02%, and insecto-acaricides paraffin oil – 770 g l-1 (89.6%) and abamectine – 18g – 1.88

19 /20

                                                                                                                                                                                                                                                                                                                                                                                                                                                                                                                                                                                                             The plantation was (THE PLANTS WERE) sprayed when
10 mobile individuals / spider mites appeared on the leaves (HOW MANY DAYS BEFORE HARVEST / approximately because of post treatment period?).
It will be good to write Tetranychus urticae in spite spider mite because it will be necessary to distinguish spider mite from two spotted spider mite

27

sodium molasses (SM)

35

Add sugar before beet also in the following rows 37,43,49,70,73, 75, 113, 153 (the sugar beet was harvested), 302

38

What is period of time; author (1) ?

70

protection of beet cultivation - protection of beet crop

72

Erysiphe betae, Cercospora beticola, Ramularia beticola, Uromyces betae, Phoma betae. Please add author

88

spider mite ? Tetranychus urticae

111

Insecticidal and acaricidal. Important is to mention both because there are insecticides that have acaricidal effect but also acaricides (mentioned in the abstract); which spider mite? (row 118 as well)

119

How many acaricides, and how many insecto-acaricides?

120

In spite TSSM please write Tetranychus urticae

121

Please explain what is optimal date.

143

No= not

149

Acaricides and insecticides

174

1 “and” is to much

173/178

Chemical control; better is expression use of plant protection products;

Row 276, 280, 283 and 290 as well; Ortus is acaricide

275

Choice = choose

276/277

There are Insecticides, insecto acaricides and acaricides

286

Write proper (insecticide, insecto acaricide, acaricide) for other active ingredients

288

Space = period

290

used to chemical treatments = used for treatments in

291

No resistance to abamectin and paraffin oil was observed. If this is explanation of the results “resistance” is not a proper expression; Better will be: The efficacy of abamectin and paraffin oil was observed.

292

Preparations = products,

297

Stadium of mites on

297/298

the influence of weather conditions, in different years, during the growing season of sugar beet

306

Spider mite =  Tetranychus urticae

308

roots. in the = roots. In the years

311

by the application of chemical control of the two spotted mite = by application of plant protection products against Tetranychus urticae

315

The type of chemical substance = Plant protection products used….

315

Conclusion : Which of the used plant protection products were not effective or were effective on Tetranychus urticae population. Target population of Tetranychus urticae was treated “x” days before harvest that post treatment period was fullfild. (Missing is something on plant protection products in the conclusion)

322

two-spotted spider mites – please use Tetranychus urticae

Author Response

Response to Reviewer 2 Comments

Reviewer #2

Point 1: Row 1-3: TITLE: Due, that the goal is to control Tetranychus urtice and influence on cultivation of sugar beet the title should be: The Effect of Acaricide Control on the Two Spotted Spider Mite Tetranychus urticae Koch in the Cultivation of Sugar Beet Beta vulgaris L. and on the Size and Quality of the Yield.

Response: We modified the title to: “The Effect of Acaricide Control on the Two Spotted Spider Mite Tetranychus urticae Koch on the Cultivation of Sugar Beet (Beta vulgaris L.) and on the Size and Quality of the Yield”.

Point 2: Row 14: In the years 2019-2021, field experiments were carried out (WHERE?).

Response: We corrected this sentence: “Field experiments (in the 2019-2021) were carried out at Department of Field Experimentation of the 115 Institute of Plant Protection – National Research Institute in Winna Góra, the purpose ...”.

Point 3: Row 15: Before beet always add “sugar” where it is missing during the text. This is for better understanding< the

purpose of which was to test the insecticidal and acaricidal effectiveness of

(SUGAR?) beet cultivation protection against T. urticae (italic) and to assess its impact on the size and quality of the (SUGAR?) beet crop.

Response: We added “sugar” before “beet”.

Point 4: Row 16: In the experiment used were acaricides:  spirodiclofen – 240 g 16 – 22.11%, mixture of hexythiazox – 250 g – 23.15% and fenpyroximate – 51.2 g – 5.02%, fenpyroximate 17 – 51.2 g – 5.02%, and insecto-acaricides paraffin oil – 770 g l-1 (89.6%) and abamectine – 18g – 1.88.

Response: We corrected this sentence.

Point 5: Row 19 /20: The plantation was (THE PLANTS WERE) sprayed when 10 mobile individuals / spider mites appeared on the leaves (HOW MANY DAYS BEFORE HARVEST / approximately because of post treatment period?). It will be good to write Tetranychus urticae in spite spider mite because it will be necessary to distinguish spider mite from two spotted spider mite.

Response: Chemical treatments were carried out in the full growing season in the phase of leaf rosette formation (July-August). The beet harvest fell in the research years in the second and third decade of October.

Point 6: Row 27: sodium molasses (SM).

Response: We added “(SM)”.

Point 7: Row 35: Add sugar before beet also in the following rows 37,43,49,70,73, 75, 113, 153 (the sugar beet was harvested), 302.

Response: We added “sugar” before “beet”.

Point 8: Row 38: What is period of time; author (1) ?

Response: 10 years: from 2010 to 2020). We added in the manuscript text: “of 2010-2020”

Point 9: Row 70: protection of beet cultivation - protection of beet crop

Response: We corrected “protection of beet cultivation” to “protection of beet crop”.

Point 10: Row 72: Erysiphe betae, Cercospora beticola, Ramularia beticola, Uromyces betae, Phoma betae. Please add author

Response: The authors are listed

Point 11: Row 88: spider mite ? Tetranychus urticae.

Response: Yes of course. Spider mite is used interchangeably to avoid repeating T. urticae.

Point 12: Row 111: Insecticidal and acaricidal. Important is to mention both because there are insecticides that have acaricidal effect but also acaricides (mentioned in the abstract); which spider mite? (row 118 as well).

Response: Has been corrected.

Point 13: Row 119: How many acaricides, and how many insecto-acaricides?

Response: Envidor, Nissorun Strong, Ortus – are acaricides; Treol and Vertigo – insecticides and acaricides.

Point 14: Row 120: In spite TSSM please write Tetranychus urticae

Response: We added “Tetranychus urticae”.

Point 15: Row 121: Please explain what is optimal date.

Response: The chemical product should be used in sugar beet from the phase of complete coverage of inter-rows until the end of root growth (BBCH 39-49), bearing in mind the grace period, which is 28 days.

Point 16: Row 143: No= not

Response: We corrected “no” to “not”.

Point 17: Row 149: Acaricides and insecticides

Response: We added “and insecticides” in this sentence.

Point 18: Row 174: 1 “and” is to much

Response: We corrected this sentence.

Point 19: Row 173/178: Chemical control; better is expression use of plant protection products;

Response: We changed “chemical controls“ to “plant protection products”.

Point 20: Row 276, 280, 283 and 290 as well; Ortus is acaricide

Response: We corrected these sentences.

Point 21: Row 275: Choice = choose

Response: We changed “choice” to “choose”.

Point 22: Row 276/277: There are Insecticides, insecto acaricides and acaricides

Response: We corrected this sentence.

Point 23: Row 286: Write proper (insecticide, insecto acaricide, acaricide) for other active ingredients

Response: We corrected this sentence.

Point 24: Row 288: Space = period

Response: We changed “space” to “period”.

Point 25: Row 290: used to chemical treatments = used for treatments in

Response: We changed “used to chemical treatments” to “used for treatments”.

Point 26: Row 291: No resistance to abamectin and paraffin oil was observed. If this is explanation of the results “resistance” is not a proper expression; Better will be: The efficacy of abamectin and paraffin oil was observed.

Response: We corrected this sentence. We changed “No resistance to abamectin and paraffin oil was observed.” to “The efficacy of abamectin and paraffin oil was observed.”.

Point 27: Row 292: Preparations = products,

Response: We changed “preparations” to “products”.

Point 28: Row 297: Stadium of mites on

Response: We corrects from “miteson” to “mites on”.

Point 29: Row 297/298: the influence of weather conditions, in different years, during the growing season of sugar beet

Response: We corrected this sentence”. Thank you very much.

Point 30: Row 306: Spider mite = Tetranychus urticae

Response: We changed “spider mite” to “Tetranychus urticae”.

Point 31: Row 308: roots. in the = roots. In the years

Response: We corrected this mistake.

Point 32: Row 311: by the application of chemical control of the two spotted mite = by application of plant protection products against Tetranychus urticae

Response: We corrected this sentence. New sentence is: “This may have been influenced by application of plant protection products against Tetranychus urticae.”.

Point 33: Row 315: The type of chemical substance = Plant protection products used….

Response: We changed “The type of chemical substance” to “Plant protection products”.

Point 34: Row 315: Conclusion : Which of the used plant protection products were not effective or were effective on Tetranychus urticae population. Target population of Tetranychus urticae was treated “x” days before harvest that post treatment period was fullfild. (Missing is something on plant protection products in the conclusion)

Response: We added text: „Vertigo 018 EC, Treol 770 EC and Ortus 05 SC were effective on Tetranychus urticae population.”.

Point 35: Row 322: two-spotted spider mites – please use Tetranychus urticae

Response: We changed “two-spotted spider mites” to “Tetranychus urticae”.
